

# Localization of potato browning resistance genes based on BSA-seq technology

Haiyan Wang[*], Ze Pang[*], Lichun Wang, Guokui Tian, Fengyun Li, Yang Pan and Kaixin Ding

[1] Keshan Branch of Heilongjiang Academy of Agricultural Sciences, Qiqihaer, Heilongjiang, China
[2] Key Laboratory of Potato Biology and Genetics, Ministry of Agriculture and Rural Affairs, Qiqihaer, Heilongjiang, China
[3] Heilongjiang Potato Germplasm Resources and Genetic Improvement Engineering Technology Center, Qiqihaer, Heilongjiang, China
[*] These authors contributed equally to this work.

## ABSTRACT

Browning is a common problem that occurs during potato processing; it is typically resolved by adding chemicals during the production process. However, there is a need to develop potato varieties that are resistant to browning due to a growing consumer interest in healthier diets. This study initially identified 275 potato varieties that are resistant to browning; these were narrowed down to eight varieties, with four of them being highly resistant. A hybrid population was developed by crossing the highly resistant CIP395109.29 with the easily browned Kexin 23. Bulked segregant analysis (BSA) was conducted, which identified 21 potato genes associated with anti-browning properties through sequencing data analysis and organization. The findings of this study lay a solid groundwork for future research on breeding potatoes with anti-browning traits, offer molecular markers for identifying anti-browning varieties, and serve as a valuable reference for further investigations into potato browning mechanisms.

## INTRODUCTION

Potatoes are the third largest food crop in the world and the fourth largest in China (*Wang et al., 2023*; *Yuhan et al., 2023*). As the largest non-cereal grain crop, it is significant in agriculture (*Li et al., 2019*), industry (*Kaur & Singh, 2016*), and other fields due to its exceptional nutritional and economic value. In recent years, as potato applications have expanded, the demand for high-quality potatoes has also increased. However, the occurrence of potato browning has become a prevalent issue during storage and processing, negatively affecting the quality, taste, and nutritional value of potatoes (*Ali et al., 2016*). Potato browning primarily occurs due to the enzymatic reaction between phenolic substances within potatoes and polyphenol oxidase (PPO) (*Vaughn, Lax & Duke, 1988*; *Yu et al., 2022*), resulting in the production of phenolic compounds. Additionally, non-enzymatic browning reactions associated with the Maillard reaction also contribute to potato browning (*Lee & Park, 2005*). These processes lead to the strengthening of

Corresponding authors
Ze Pang, pangze96@163.com
Lichun Wang, wanglichun78@163.com

pigments, browning, and the formation of unsightly spots. Browning frequently occurs in potato processing industries, significantly impacting food production, food safety, and environmental pollution, resulting in avoidable losses. Therefore, preventive measures must be taken (*Vámos-Vigyázó, 1981*; *McEvily, Iyengar & Otwell, 1992*).

Eliminating or delaying potato browning is a significant concern in product processing and home applications. Common methods to prevent browning include physical and chemical approaches. Physical methods encompass low-temperature preservation, low-frequency ultrasound (*Xu et al., 2022*), vacuum and modified atmosphere preservation (*Rocculi et al., 2009*). Chemical methods mainly involve a preservative food coating (*Shun-Shun, 2010*) or the use of anti-browning agents (*Nascimento et al., 2020*; *Ru et al., 2020*). In the processing industry, the most prevalent method for preventing potato browning is the use of anti-browning agents. These agents effectively and simply slow down the occurrence of potato browning. Sulfite is an efficient browning inhibitor, and potato producers control browning by applying sulfite. But with widespread application, adverse factors have also been exposed, which may have adverse effects on physical health. Therefore, the US Food and Drug Administration has restricted the use of sulfites. As a result, various substitutes are being sold on the market, such as ascorbic acid or a combination of isoascorbic acid with citric acid and cysteine. *Nicolau-Lapea et al. (2021)* studied the anti-browning treatment of sliced potatoes using two antioxidant solutions, and demonstrated the effectiveness of vitamin C as a substitute for sulfite in preserving fresh cut potatoes and delaying their browning. However, these products are oxidized irreversibly and therefore do not meet the shelf life requirements in pre-peeled potatoes without special packaging or cover solutions (*Sapers & Miller, 1993*). The limitations of some of the anti-browning agents and the pressure from regulatory agents point to the need for developing alternative technologies for the prevention of enzymatic browning that will be effective and safe. Therefore, solving the potato browning problem from a genetic perspective is the most fundamental approach.

Bulked segregant analysis (BSA) was performed by selecting parents with relative traits to construct a separate population. We chose a specific number of individuals with extreme phenotypes to form a mixed pool and used the differences in DNA molecular markers between the two mixed pools to achieve quantitative trait locus (QTL). BSA, when combined with high-throughput genome resequencing technology, offers an efficient and accurate way to identify trait determining genes, while also reducing research costs. This technology has found widespread application in various animal and plant breeding research, with promising prospects for future applications. For example, *Qiu et al. (2022)* successfully used BSA-seq technology to locate the carpel number of watermelon and identified a 44.7 Mb chromosome fragment related to the carpel number trait on chromosome 7. The candidate gene Cla97C07G143260 was screened. *Chen et al. (2021)* conducted research on the slow melting flush (SMF), and BSA results identified two adjacent main QTLs on chromosome 4, resulting in a total of 29 genes that may be related to SMF. *Yang et al. (2021)* obtained the candidate gene CsaV3_ 6G050410 for controlling trichomes through BSA-seq technology. *Li et al. (2021)* used BSA-seq technology to map potato starch traits, located them on chromosome 2, and obtained six candidate genes. *Sharma et al. (2021)* studied the dormancy and germination of potato tubers. At present, there is relatively

little research on the application of BSA seq technology in potatoes, especially in potato browning. Therefore, our research is particularly significant. This study aimed to evaluate the degree of browning in 275 potato samples, create hybrid combinations, and use BSA-seq technology to identify candidate genes associated with potato browning. The findings of this study will provide a theoretical basis for future screening of anti-browning materials in potatoes.

## MATERIAL AND METHODS

### Experimental materials and identification
#### Identification of potato resources browning
This study was conducted at the Keshan branch of the Heilongjiang Academy of Agricultural Sciences from 2018 to 2019 and relied on platforms such as the National Potato Improvement Center, the National Potato Germplasm *in vitro* Seedling Bank (Keshan), and the Key Laboratory of Potato Biology and Genetic Breeding of the Ministry of Agriculture. A total of 275 potato varieties were used (Table S1). We then identified how each variety browns through indicators such as browning index, browning intensity, and cooking browning intensity.

#### Construction of offspring population and identification of browning
Based on the results of our browning identification research, the anti-browning material CIP395109.29 was selected as the female parent, and the low resistance browning material KX 23 was selected as the male parent. A hybrid F1 population containing 362 families was constructed. From 2021 to 2022, planting and sowing were conducted on the experimental site of Keshan Branch of Heilongjiang Academy of Agricultural Sciences. After identifying the degree of browning in the offspring population, 30 light browning and 30 heavy browning potato resources were selected for mixed pool construction.

### Experimental methods
#### Browning index
The methods developed by *Wang et al. (2007)* were used to determine the browning index. We selected potato tubers of uniform size, with no pests or diseases, and no green skin. Three samples of each potato were selected with two tubers per repeat. The tubers were cut evenly and then sectioned into 0.5 cm thick potato chips. The samples were photographed every 30 min, 2 h, 5 h, and 7 h after cutting. Classification of browning levels based on the browning area on the cutting surface (Table S1):

Note: The browning index is calculated according to browning grading standards. The higher the browning index, the heavier the degree of browning.

$$\text{Browning Index} = \Sigma \left[ \frac{\text{level} \times \text{tubers number of this index}}{\text{top level} \times \text{all of tubers number}} \right] \times 100\%.$$

#### Browning strength
We referred to *Li et al. (2010)* to determine the intensity of browning. After peeling the potato, 3 mm of subcutaneous flesh was removed and a sample was taken from the surface

of the tuber for skin samples. Heart samples were obtained approximately 3 cm from the center of the potato tuber. We weighed 1 g of the sample, chopped it, added deionized water at a ratio of 1:4 (m/m), homogenized it with a high-speed tissue grinder for 5 min, and placed it in a 30 °C water bath for 20 min. We then took a portion of the homogenization at 4 °C, centrifuged it at 12,000 r/min for 5 min, and removed the supernatant to measure the absorbance value at 420 nm. The remaining homogenate was kept at 4 °C for 24 h and centrifuged at 12,000 r/min for 5 min; the resulting supernatant was taken to measure the absorbance value at 420 nm. Using distilled water as the blank control, each treatment was repeated three times and parallel samples were obtained. The results are calculated as the average value, and A420 was the browning intensity (BD). Here, materials with a core browning strength less than 0.25 after 20 min of browning at 30 °C and a change value of no more than 0.15 after being placed at 4 °C for 24 h were selected as anti-browning materials due to the lack of a unified standard for browning strength.

### Steam-cooked browning

We referred to *Bradshaw (2013)* to determine cooked browning. Three potato tubers with no mechanical damage, no pests or diseases, no green skin, and moderate size were selected as samples for testing. The potatoes were peeled and washed and were kept under the water until cooking to avoid enzymatic discoloration. The samples were placed in an electric rice cooker with just enough water to reach the tubers. The tubers were cooked for 25 min, checked for doneness and the undercooked pieces were cooked for another 5 min. The steamed tubers were left under natural conditions for 24 h and their discoloration was evaluated. A time delay may deepen the discoloration and improve ability to evaluate any differences. According to Dutch breeders, the degree of browning after cooking can be categorized into six levels: 9, 8, 7, 6, 5, and 4. The 9 represents no discoloration, and 4 represents dark gray or black. The color of the steamed potato tubers was used in conjunction with this standard to determine the grade.

### Mixed pool construction and DNA extraction

To construct the extreme material pools, 30 families exhibiting extreme anti-browning and 30 families with extreme low anti-browning were selected from the F1 generation segregation population, along with their male and female parents. DNA was extracted from these samples using the CTAB method and stored at −20 °C.

### Library construction and sequencing

The reference genome was selected from the *S. tuberosum* group phureja DM 1−3 V 6.1 (http://spuddb.uga.edu/). Each offspring in the mixed pool has a sequencing depth of ≥ 1 x, and each parent has a sequencing depth of ≥ 20 x. The detection of SNP and InDel was mainly achieved using the GATK software toolkit. The specific process was determined by the best practices found on the GATK official website (https://www.broadinstitute.org/gatk/guide/best-practices?bpm=DNAseq#variant-discovery-ovw). The data was then filtered and analyzed by the Beijing Biomarker Technologies Co. Resequencing was used to obtain two extreme trait pools of potato, one with high browning (Z1) and the other with low browning (Q1), as well as genotype data of the two parental parents. The extreme pool

was derived from the F1 family population, and a BSA analysis strategy was used to locate trait-related loci. The Euclidean Distance (ED) method was employed for localization analysis.

$$ED = \sqrt{(A_{mut} - A_{wt})^2 + (C_{mut} - C_{wt})^2 + (G_{mut} - G_{wt})^2 + (T_{mut} - T_{wt})^2}$$

where $A_{mut}$ is the frequency of A base in the mutation pool, $A_{wt}$ is the frequency of A base in the wild-type pool; $C_{mut}$ is the frequency of C bases in the mutation pool, and $C_{wt}$ is the frequency of C bases in the wild-type pool; $G_{mut}$ is the frequency of G bases in the mutation pool, and $G_{wt}$ is the frequency of G bases in the wild-type pool; $T_{mut}$ is the frequency of T bases in the mutation pool, and $T_{wt}$ is the frequency of T bases in the wild-type pool.

The larger the ED value, the greater the difference between the two mixing pools.

## RESULTS

### Evaluation of browning in 275 potato resources

A comprehensive evaluation was conducted on 275 potato samples, assessing their browning index, browning intensity, and cooking browning index. The results revealed a wide variation in the anti-browning abilities of the potatoes. From this evaluation, eight anti-browning samples were selected, including four high anti-browning materials, namely Yunshu 501, CIP395109.29, CIP393615.6, and Yunshu 401 (Table 1).

### Formation of extreme bulks and sequencing data quality control

Following the identification of the varieties used to determine potato browning, Kexin 23, was identified as a susceptible browning resource and was chosen as the male parent. Sexual hybridization was conducted between Kexin 23 and the four potato varieties that were highly resistant to browning. During the evaluation, it was observed that Yunshu 401 did not exhibit typical fruiting behavior, while CIP395109.29 displayed a high fruit setting rate. Consequently, we selected CIP395109.29 as the female parent for sexual hybridization in the population construction process. A total of 4,922 actual seeds were obtained. In 2020, 700 seeds were randomly selected and treated with 1.5 mg/ml gibberellin, resulting in a total of 421 actual seedlings. From 2021 to 2022, the degree of browning of the F1 population was identified. Thirty individuals with milder browning and 30 individuals with heavier browning were selected from the offspring population to construct a mixed pool, along with the two parents (Table S3). The sample concentration was ≥ 30 ng/ μl, the total sample quantity was ≥ 2 μg (not less than 15 μl), and the sample purity OD260/280 was 1.6−2.5.

The results indicated that the GC content ranged between 34.58% and 36.29%, the genome coverage depth was greater than 95% (with at least three base covers), and Q30 was higher than 93%. These parameters confirm that the sequencing sample data was sufficient, the sequencing data quality is qualified, and it aligns with the potato reference genome (http://spuddb.uga.edu/dm_v6_1_download.shtml). The comparison efficiency is high, indicating its suitability for subsequent mutation detection and gene localization of traits (Table 2)

**Table 1  Screening and identification of anti browning potato resources.**

| Resource name | Browning index | Browning intensity | | Cooking browning index |
|---|---|---|---|---|
| | | 30 °C 20 min | 4 °C 24 h | |
| Yunshu 501 | 0 | 0.117 | 0.245 | 9 |
| CIP395109.29 | 0 | 0.185 | 0.255 | 7 |
| CIP393615.6 | 0 | 0.151 | 0.180 | 5 |
| Yunshu 401 | 12.50 | 0.112 | 0.133 | 8 |
| NH High starch | 25.00 | 0.089 | 0.121 | 8 |
| CIP397100.9 | 29.17 | 0.134 | 0.159 | 9 |
| K200950-3 | 33.33 | 0.154 | 0.210 | 9 |
| S.goniocalyx | 37.50 | 0.121 | 0.158 | 8 |

**Table 2  Quality inspection of mixing pool.**

| Sample | Total reads | Comparison rate between sequencing data of each sample and genome (%) | Genome coverage depth (>3) | Alkali base mass value greater than 30 |
|---|---|---|---|---|
| fuben | 127075230 | 87.68 | 95 | 94.14 |
| muben | 146439062 | 78.71 | 95 | 94.04 |
| Q1 | 375407052 | 88.64 | 98 | 93.64 |
| Z1 | 376789854 | 88.91 | 98 | 93.42 |

## The distribution of browning sites on chromosomes

Following the filtering of SNPs and InDels, a total of 14,339,422 original SNPs and InDel sites were obtained. Subsequent screening resulted in 7,910,511 high-quality and reliable SNPs and InDel sites (Table 3 and Fig. 1). The ED values of each point were calculated and multiplied, and the 5th power of the original ED value was taken as the correlation value to eliminate background noise. The LOESS method was employed to fit the ED values, and the correlation threshold was set at the median + 5 sd of the fitted values of all sites. This analysis revealed a single interval associated with the region at 24.08–27.10 Mb on chromosome 8, where genes related to controlling potato browning were identified (Fig. 2).

## Functional annotation of associated interval genes

Utilizing the localization region, gene location, and reference genome information, potato browning-related genes were mapped onto chromosome 8, resulting in the annotation of a total of 21 candidate genes. Functional annotation of these candidate genes was carried out using seven functional databases, including NR, TrEMBL, KEGG, GO, KOG, SwissProt, and PFAM. The functions and annotations of each gene are detailed in the Table S1. For detailed results, please refer to Data S1.

**Table 3  The distribution of high-quality credible loci on chromosomes.**

| chr | SNP_number | InDel_number | Variants_number |
|---|---|---|---|
| chr01 | 828616 | 134189 | 962805 |
| chr02 | 439850 | 88117 | 527967 |
| chr03 | 568264 | 93970 | 662234 |
| chr04 | 747313 | 110071 | 857384 |
| chr05 | 576186 | 91108 | 667294 |
| chr06 | 575172 | 93196 | 668368 |
| chr07 | 631002 | 95277 | 726279 |
| chr08 | 435291 | 79120 | 514411 |
| chr09 | 648282 | 89206 | 737488 |
| chr10 | 403903 | 53818 | 457721 |
| chr11 | 552832 | 82504 | 635336 |
| chr12 | 426189 | 67035 | 493224 |
| total | 6832900 | 1077611 | 7910511 |

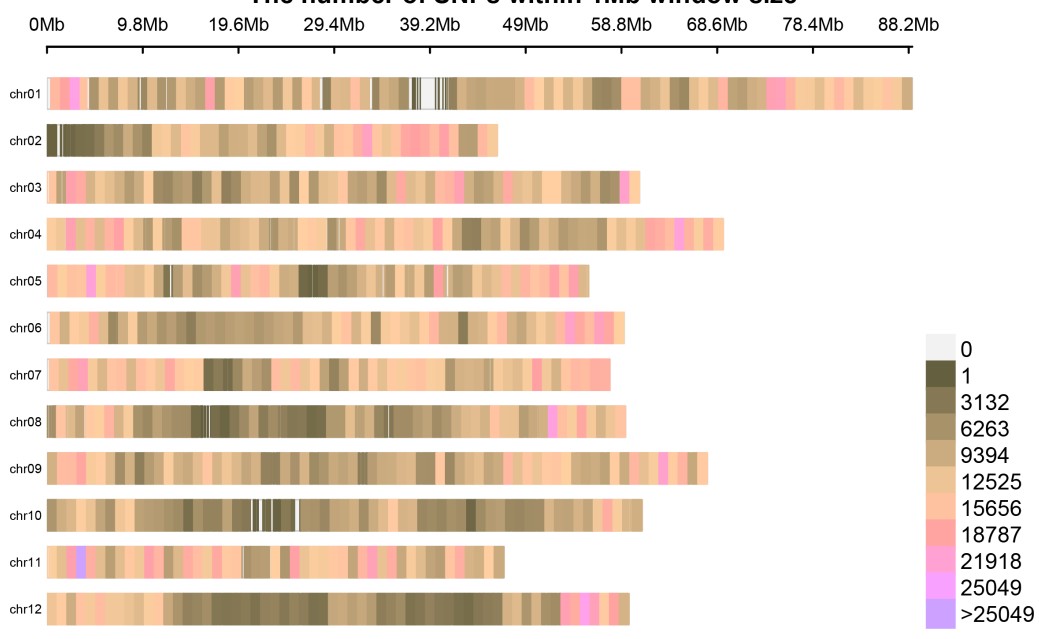

**Figure 1  The distribution of high-quality credible loci on chromosomes.**

## DISCUSSION

### Screening for potato resistance to browning: a vital step in browning resistance breeding

Traditionally, inhibiting potato browning has heavily relied on the use of anti-browning chemical reagents during processing, which has proven in the past to be the most effective method. However, with the growing emphasis on health concerns, there is a rising demand
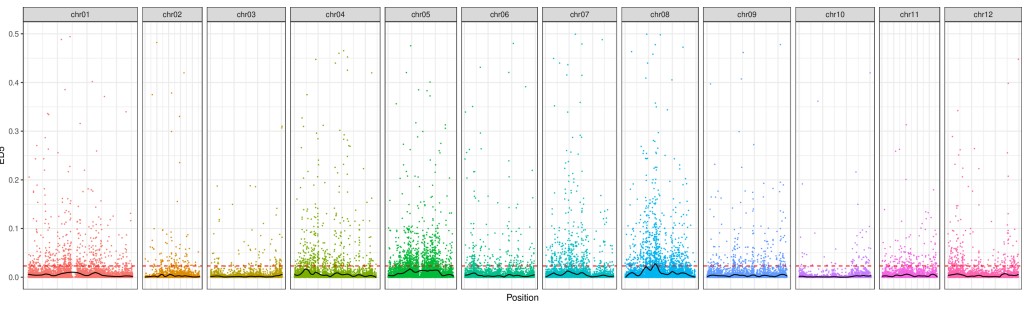

**Figure 2** **The distribution of ED correlation values on chromosomes.**

for green, healthy, and environmentally-friendly processed products (*Lam et al., 2013*). Consequently, enhancing the natural ability of potatoes to resist browning has become increasingly important. We used 275 potato samples for anti-browning identification, resulting in the identification of eight anti-browning varieties, including four highly effective anti-browning potatoes. A total of 421 offspring resources were obtained through the population configuration, with the highly resistant browning resource CIP395109.29 serving as the female parent and the easily browning resource Kexin 23 as the male parent. After two years of browning degree identification, 30 anti-browning and 30 susceptible browning resources were selected. Given that potatoes are not native to our country, resources are relatively limited. Typically, chemical agents are the primary method used to prevent browning during processing. Sulfite, a commonly used additive, is widely employed in the potato processing industry. However, sulfite is an allergen with potential adverse effects on human health. Additionally, studies have suggested that vitamin C can be utilized as an anti-browning substance in products. While these methods can effectively alleviate or mitigate the browning issue during potato processing, they may negatively impact processing quality and increase processing costs, ultimately hindering the development of the potato processing industry. Improving variety defects through breeding methods has consistently been the most widely used and effective approach in agriculture. Therefore, enhancing the natural resistance of potatoes to browning is the most direct, green, and environmentally friendly solution to address processing browning concerns.

## Screening of potato browning resistance related genes

Through the tireless efforts of scientific researchers, genes related to potato traits have been continuously discovered and emerged. Of particular interest are genes related to brown traits, as demonstrated by *Bachem et al. (1994)*. Who showed that the expression of antisense RNA from a potato PPO cDNA can decrease PPO activity and inhibit enzymatic browning in European potato varieties selected for blackspot resistance. Similarly, *Coetzer et al. (2001)* demonstrated that sense and antisense RNA from a heterologous tomato PPO gene can control the level of PPO activity and enzymatic browning in the major commercial potato cultivar in the United States, the Russet Burbank potato. Enzymatic browning mediated by PPO is particularly evident in potato tubers (*Matheis, 1987*; *Corsini, Pavek & Dean, 1992*), where the enzyme is localized within amyloplasts of the tuber cells

(*Mayer & Harel, 1979*). Two PPO genes previously isolated from potato by Hunt et al. were found to be expressed in leaves, flowers, roots, and petioles, but no expression was detected in tubers (*Hunt et al., 1993*). In a recent study, BSA technology was utilized to locate browning-related genes on chromosome 8, identifying a total of 21 candidate genes that may be involved in the browning process. Therefore, further in-depth research into these genes holds significant importance.

Research has revealed that 08G009100 and 08G009110 are potentially associated with phosphoinositol phosphatase SAC9, while 08G009160 is speculated to be a gene related to the F-box protein family. Additionally, 08G009210 may be linked to methylthioribose-1-phosphate isomerase, and 08G009230, 08G009370, and 08G009420 may translate proteins containing pentapeptide repeat sequences that are potentially related to the RPF2, TPR, and PPR families, respectively. Meanwhile, 08G009280 and 08G009290 are speculated to be associated with chlorophyll (ide) b reductase NOL, and 08G009310 and 08G009320 may have the potential 3-hydroxybutyryl CoA hydrolase 3 function. The function of 08G009400 may be related to the WAT1 protein, while 08G009450 may be related to ADP ribosylation factor GTPase activating protein AGD2 like isoform X1. From this, it is clear that potato browning is a complex physiological process that requires multiple genes to cooperate and regulate in order to have an anti-browning effect. Further screening and investigation are required to determine the primary functional roles of these 21 candidate genes.

## CONCLUSION

We have successfully identified the browning characteristics of 275 potato samples, leading to the discovery of high-quality anti-browning varieties. This breakthrough serves as a solid foundation for future advancements in breeding potatoes that do not brown. Notably, we have pioneered the use of BSA technology to pinpoint the anti-browning related genes on chromosome 8 within the 24.08–27.10 Mb range, resulting in the identification of 21 candidate genes. This groundbreaking revelation lays the groundwork for further in-depth exploration of potato browning. Building upon these pivotal findings, our subsequent research aims to uncover even more compelling insights.

In summary, the experimental research results will hopefully serve as useful feedback information to improve browning resistance in potatoes. These results may catalyze advancements in the potato industry and improve food security measures.

### Funding

This research was supported by the Genetic and QTL mapping analysis of potato browning (2020YYYF004), the China Potato Industry Technology System Qiqihar Comprehensive Test Station (CARS-09-ES37), the Research on the Breeding of Breakthrough New Varieties of Fresh Potatoes (CX23GG02), the Selection of Green, High Quality and Efficient Potato Varieties (2021ZXJ05A0503-2), the Experimental Demonstration of Green and Smart Agriculture Achievements in Potatoes (2021ZXJ05A0503-4), the Research on the

Creation of High Quality and Multi resistant Processing Type New Potato Germplasm (2022ZXJ06B02-02a), the Innovation and utilization of potato early maturing high-quality multi resistance germplasm resources (2022ZXJ06B01-03), and the Mining and Expression Analysis of Thaxtomin A Tolerance Genes in Potato Scab Virus (CX22YQ30). The funders had no role in study design, data collection and analysis, decision to publish, or preparation of the manuscript.

## Grant Disclosures

The following grant information was disclosed by the authors:
The Genetic and QTL mapping analysis of potato browning: 2020YYYF004.
The China Potato Industry Technology System Qiqihar Comprehensive Test Station: 2020YYYF004.
The Research on the Breeding of Breakthrough New Varieties of Fresh Potatoes: CX23GG02.
The Selection of Green, High Quality and Efficient Potato Varieties: 2021ZXJ05A0503-2.
The Experimental Demonstration of Green and Smart Agriculture Achievements in Potatoes: 2021ZXJ05A0503-4.
The Research on the Creation of High Quality and Multi resistant Processing Type New Potato Germplasm: 2022ZXJ06B02-02a.
The Innovation and utilization of potato early maturing high-quality multi resistance germplasm resources: 2022ZXJ06B01-03.
The Mining and Expression Analysis of Thaxtomin A Tolerance Genes in Potato Scab Virus: CX22YQ30.

## Competing Interests

The authors declare there are no competing interests.

## Author Contributions

- Haiyan Wang conceived and designed the experiments, performed the experiments, analyzed the data, prepared figures and/or tables, authored or reviewed drafts of the article, and approved the final draft.
- Ze Pang performed the experiments, analyzed the data, prepared figures and/or tables, authored or reviewed drafts of the article, and approved the final draft.
- Lichun Wang conceived and designed the experiments, performed the experiments, authored or reviewed drafts of the article, and approved the final draft.
- Guokui Tian performed the experiments, authored or reviewed drafts of the article, and approved the final draft.
- Fengyun Li performed the experiments, authored or reviewed drafts of the article, and approved the final draft.
- Yang Pan performed the experiments, authored or reviewed drafts of the article, and approved the final draft.
- Kaixin Ding performed the experiments, authored or reviewed drafts of the article, and approved the final draft.

# PeerJ

## Data Availability

PRJNA1102823.

## Supplemental Information

Supplemental information for this article can be found online at http://dx.doi.org/10.7717/peerj.17831#supplemental-information.

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
