# Peer review of "Localization of potato browning resistance genes based on BSA-seq technology"

_PeerJ, doi:10.7717/peerj.17831_

## Round 0.1 · original submission · Major Revisions

Three reviewers have critical comments demanding major revision. We have even a suggestion to reject the manuscript. However, I believe this novel work could revised. Please consider all the remarks, and check the attached annotated manuscript kindly provided by reviewer #2. Take more time for revision, if necessary. Please check English.

**Language Note:** The Academic Editor has identified that the English language must be improved. PeerJ can provide language editing services - please contact us at [email protected] for pricing (be sure to provide your manuscript number and title). Alternatively, you should make your own arrangements to improve the language quality and provide details in your response letter. – PeerJ Staff

Reviewer 1 ·

Basic reporting

The writing may be improved. Little description of results section.

Experimental design

no comment

Validity of the findings

The identified anti-browning potato varieties are useful for breeding. The candidate gene may be determined by more analysis.

Additional comments

Wang et al evaluated the 275 browning characteristics of 275 potato resources and identified a QTL with 21 candidate genes. This work is generally important for potato anti-browning breeding. I have a few concerns regarding presentation of the data and its interpretation that should be addressed in a major revised manuscript.

Major comments:

1. In the introduction section, there is little description of potato browning or research progress in genetics. It would be better to add some introduction related to BSA-Seq progress in potato and simplify the definition of the BSA-Seq method.

2. In the method section of Library Construction and Sequencing, no introduction of the sequencing is provided, such as the reference genome, the depth of sequencing, etc.

3. How about presenting the BSA-Seq results by ΔSNP index?

4. In the results section, more information about the results should be described. For example, in the 21 identified genes of the interval, what are the possible candidates based on sequence variation and gene expression analysis, or reported functional identification in other plants? Also, in lines 193-197, where is the functional database annotation of NR, TrEMBL, KEGG, GO, KOG, SwissProt, and PFAM in table 7?

5. Provide evaluation information for all potato resources in table 1 or 3. Additionally, why was CIP393615.6 not chosen as the anti-browning potato resource?

Minor comments:

1. There is a question regarding potatoes being highly heterozygous autotetraploids, and whether the ED calculation method described in the article can accurately identify candidate loci. First, is resistance to browning a dominant or recessive trait? If dominant, then the SNP frequency in the dominant pool cannot be accurately calculated.

2. Ensure that the number of F1 is properly subscripted and provide figure legends.

3. Check for grammar mistakes.

Reviewer 2 ·

Basic reporting

Simple a little

Experimental design

not perfectly

Validity of the findings

There are some findings.

Additional comments

This study identified potato resources with resistance to browning, selected some with high resistance. However, Molecular markers for identifying resistant varieties are comparatively weaker.

Annotated reviews are not available for download in order to protect the identity of reviewers who chose to remain anonymous.

Reviewer 3 ·

Basic reporting

This is basic research that provides a basis for exploring the browning resistance genes in potatoes. However, this manuscript needs to be improved before publication with the following issues/concerns resolved.
1. The abstract written is too general. Need to highlight the results with follow-up.
2. Cite the reference for each protocol followed in the study.
3. The methods for 'Library Construction and Sequencing' need more in detail. e.g Data filtration, SNP calling etc. What criteria were used to identify the uniquely mapped reads?
4. Line 168- Group Construction should be replaced by the 'formation of extreme bulks'
5. Before outsourcing for sequencing, how much quantity of DNA of each sample is used for preparing the bulks?
6. In Table 4-Browning weight should be rewritten as browning dark
7. Lines 185-186 'Subsequent screening resulted in 7,910,511 high-quality and reliable SNPs and
186 InDel sites (Table 6 and Fig. 1)'. What it refers to?
8. Line 183- heading is not appropriate.
9. Line 254-Through extensive discussions, -Discussion here is not appropriate, authors can write-investigation.
10. Why did the authors not use the interval parameter and sliding window approach for delta SNP-index calculation as in Takagi et al., 2013, and instead used the ED approach?
11. A total of 21 putative candidate genes within 3.0 Mb interval were identified in this study, which should be verified by RT-PCR experiment. Meanwhile, the published RNA-seq is suggested to be cited in this manuscript to investigate the expression pattern in response to browning, and qRT-PCR experiments helps to confirm the key browning resistance candidate genes. If the authors add these relative results, the manuscript will be more integral to provide significantly valuable information for other researchers.
12. Have you submitted the raw data to NCBI?

Experimental design

No comments

Validity of the findings

The results need to be validated. Further, experiments to be designed and follow-up may be highlighted in the conclusion section.

---

## Round 0.2 · Minor Revisions

We have several positive comments from the reviewers. However, the reviews have some critical remarks. Check references formatting. Please consider detailed comments by reviewer #2.

Reviewer 1 ·

Basic reporting

general.

Experimental design

general.

Validity of the findings

This study initially screened 275 potato varieties for browning resistance, with four displaying high resistance. Bulked segregant analysis (BSA) identified 21 genes linked to anti-browning traits through BSA-sequencing data analysis.

Additional comments

This version is better in terms of text description. I have no concerns. There are still many grammar errors in the text and should be addressed.

Reviewer 2 ·

Basic reporting

Browning during processing is a common issue with potatoes, and the usual solution involves adding chemicals during production.

Experimental design

No problem

Validity of the findings

This study initially identified 275 potato resources with resistance to browning and then narrowed it down to 8 resistant potato resources. Subsequently, Bulked Segregant Analysis (BSA) was conducted on the population, leading to the identification of 21 potato genes associated with anti-browning properties through sequencing data analysis and organization.

Additional comments

There are a few tables (1, 4, 7) that are too large, and it is advised to show them as supplementary tables;
Table 2 is a list of browning degrees, and it is recommended to display it as a supplementary table 2;
The author has analyzed by different means, including NR, TrEMBL, KEGG, GO, KOG, SwissProt, and PFAM. Here we need add some relevant results (figures);
Some references have no volumes or pages; The title in the references, except for the first letter capitalization, the others need to be lowercase; Please make uniform corrections to the reference format according to the requirements of the journal.

Annotated reviews are not available for download in order to protect the identity of reviewers who chose to remain anonymous.

Reviewer 3 ·

Basic reporting

The authors appear to have addressed most of the concerns raised by the reviewers of the original manuscript. However, some grammar issues still need work in a few places, and minor scientific language editing is required.

Experimental design

Appropriate and well-explained.

Validity of the findings

Validation of candidate gene(s) requires additional experiments which could become another publication.

Additional comments

Nil

---

## Round 0.3 · Minor Revisions

The manuscript got better after the revision. However, minor remarks remain. Please check the comments kindly provided by reviewer #2 in the PDF file. Waiting resubmission of the revised manuscript.

Reviewer 1 ·

Basic reporting

General.

Experimental design

General.

Validity of the findings

General.

Additional comments

I have no concerns for this revised version. it may be considered for acceptable.

Reviewer 2 ·

Basic reporting

The manuscript has been improved after two revisions, but there are still some problems with the manuscript. Suggested Major revision;
When there are two or more references, please order them in the order of publication time and author's name. For example, Line 50,253,
The authors still pay attention to spaces (or blank) in the text, such as Line 211,278; 316;
There are too few pictures in the manuscript, so it is recommended to add a few Figs about KEGG, GO and other analyses to improve the quality and content of the manuscript;
The title of the references should be lowercase, except for the capitalization of the first letter; In addition, the Latin name of the plant species inside needs to be italicized;
Table 7 is also too large, and it is difficult to publish within a single page, so it is recommended to list it as a supplementary Table.

Experimental design

No problem

Validity of the findings

No problem

Additional comments

No problem

Annotated reviews are not available for download in order to protect the identity of reviewers who chose to remain anonymous.

---

## Round 0.4 · accepted · Accept

Thank you for the manuscript update and detailed reply to the reviewers. The paper will be published soon. Only some technical remarks remain to be fixed at proofreading stage. Please check the remarks in the annotated PDF file kindly provided by reviewer #2.


Reviewer 1 ·

Basic reporting

General.

Experimental design

General.

Validity of the findings

General.

Additional comments

I have no concerns with this revised version, and it is suitable for acceptance.

Reviewer 2 ·

Basic reporting

The authors have made corrections to the manuscript as required, but there are still some corrections that need to be made in terms of references, and the journal name, volume and page number of the relevant references should be added.
The reviewer has marked it in the PDF file.

Experimental design

It is good.

Validity of the findings

well

Additional comments

Well

Annotated reviews are not available for download in order to protect the identity of reviewers who chose to remain anonymous.